# Combination of Symptom Profile, Endoscopic Findings, and Esophageal Mucosal Histopathology Helps to Differentiate Achalasia from Refractory Gastroesophageal Reflux Disease

**DOI:** 10.3390/diagnostics11122347

**Published:** 2021-12-13

**Authors:** Chia-Chu Yeh, Chia-Tung Shun, Liang-Wei Tseng, Tsung-Hsien Chiang, Jia-Feng Wu, Hui-Chuan Lee, Chien-Chuan Chen, Hsiu-Po Wang, Ming-Shiang Wu, Ping-Huei Tseng

**Affiliations:** 1Department of Internal Medicine, National Taiwan University Hospital, Taipei 100, Taiwan; smallwater718@gmail.com (C.-C.Y.); orangewell@gmail.com (L.-W.T.); chienchuanchen@ntu.edu.tw (C.-C.C.); wanghp@ntu.edu.tw (H.-P.W.); mingshiang@ntu.edu.tw (M.-S.W.); 2Good Liver Clinic, Taipei 100, Taiwan; ctshun@ntu.edu.tw; 3Department of Pathology, National Taiwan University Hospital, Taipei 100, Taiwan; 4Department and Graduate Institute of Forensic Medicine, College of Medicine, National Taiwan University, Taipei 100, Taiwan; 5Division of Gastroenterology and Hepatology, Department of Internal Medicine, Fu Jen Catholic University Hospital, Fu Jen Catholic University, New Taipei City 24352, Taiwan; 6Department of Integrated Diagnostics and Therapeutics, National Taiwan University Hospital, Taipei 100, Taiwan; thchiang1118@yahoo.com.tw; 7Department of Pediatrics, National Taiwan University Children’s Hospital, Taipei 100, Taiwan; wjf@ntu.edu.tw (J.-F.W.); huijuan622@gmail.com (H.-C.L.)

**Keywords:** achalasia, endoscopic biopsy, gastroesophageal reflux disease, high-resolution impedance manometry, mucosal histopathology

## Abstract

Achalasia, a rare primary esophageal motility disorder, is often misdiagnosed as refractory gastroesophageal reflux disease (GERD). This study is aimed to identify the clinical and histopathologic features that may help to differentiate these two entities. Patients with untreated achalasia and those with refractory reflux symptoms despite ≥8 weeks of proton-pump inhibitor treatment were enrolled prospectively. All patients underwent validated symptom questionnaires, esophagogastroduodenoscopy with biopsy, and high-resolution impedance manometry (HRIM). Histopathology of esophageal mucosa was compared based on four pre-determined histological criteria: basal cell hyperplasia or papillae elongation, eosinophilic infiltration, petechiae formation, and hypertrophy of the muscularis mucosae (MM). Compared with the GERD patients, patients with achalasia had similar reflux symptoms, but higher Eckardt scores, fewer erosive esophagitis and hiatal hernia, more esophageal food retention on endoscopy, and higher prevalence of hypertrophy of the MM and petechiae formation on histopathology. Multivariate logistic regression based on Eckardt score ≥4, normal esophagogastric junction morphology or esophageal food retention, and coexistence of petechiae formation and hypertrophy of the MM, established the best prediction model for achalasia. Therefore, combination of routinely accessible variables, including Eckardt score, endoscopic features, and histopathology obtained via esophageal mucosal biopsy, may provide an earlier identification of achalasia.

## 1. Introduction

Achalasia is a rare primary esophageal motility disorder involving the smooth muscle of the esophageal body and the lower esophageal sphincter (LES). The annual incidence is approximately 1.6 cases per 100,000 individuals [1]. In patients with non-obstructive dysphagia, achalasia is the most common etiology [2,3]. Patients suffer from progressive dysphagia to both solids and liquids, resulting in significant weight loss. Currently, high-resolution manometry is the gold standard for an accurate diagnosis of achalasia [4]. Treatment choices for achalasia includes medical treatment, surgical myotomy, and endoscopic treatment, such as pneumatic dilatation, botulinum toxin injection, and peroral endoscopic myotomy (POEM). Among them, POEM, a minimally invasive, safe, and effective endoscopic procedure, has become a popular first-line treatment modality worldwide [5,6]. Nevertheless, the diagnosis of achalasia is often delayed due to its insidious onset and the gradual progression of the disease that can occur over several years, as well as the non-specific symptoms, such as regurgitation, chest pain, and heartburn, that mimic gastroesophageal reflux disease (GERD) [7]. Without an early diagnosis and an adequate treatment, achalasia may progress to a sigmoid-type esophagus with a markedly tortuous and dilated esophageal lumen, requiring an esophagectomy and surgical reconstruction [8]. In addition, these patients have a higher risk of developing esophageal cancer, predominantly squamous cell carcinoma [9].

Achalasia is most frequently misdiagnosed as GERD [10]. Therefore, it is imperative to find useful and feasible tools to differentiate achalasia from GERD. For patients with initial symptoms of dysphagia, acid regurgitation, or heartburn, an esophagogastroduodenoscopy (EGD) is usually the first diagnostic modality to evaluate the presence of erosive esophagitis and to exclude mechanical causes, including peptic stricture or malignant obstruction. Nevertheless, the sensitivity of EGD only for detecting early achalasia without a dilated esophageal lumen is poor. Retained food in the dilated esophagus was fermented by bacteria into lactic acid and created an acidic environment. Previous studies reported that untreated achalasia patients also experienced acid reflux by 24-h pH monitoring and presented with erosive esophagitis by EGD examination, making the differential diagnosis form refractory GERD more difficult [11]. Currently, in patients with GERD, microscopic evidence of esophagitis caused by reflux has been suggested to provide adjunctive evidence for the diagnosis of GERD [12,13]. In patients with achalasia, impaired relaxation of the LES and aperistalsis cause liquid and solid retention in the esophageal lumen, ultimately leading to mucosal inflammation, which may be observed endoscopically or microscopically. Although biopsies of the esophageal mucosa are readily available during EGD, only a few studies have focused on the histopathological characteristics of the esophageal mucosa in achalasia [14]. Therefore, in the present study, we aimed to investigate whether the histopathological characteristics of the esophageal mucosa in patients with achalasia obtained via routine endoscopic biopsies may help differentiate achalasia from GERD [10]. Furthermore, we aimed to investigate whether the combination of clinical, endoscopic, and histopathologic characteristics can improve the diagnostic performance of achalasia, avoiding misdiagnosis and disease progression [15].

## 2. Methods

### 2.1. Study Design and Patients

This study was conducted at a tertiary medical center with approval from the Research Ethics Committee. All participants gave their written informed consent before participating in the study. Patients who were newly diagnosed with achalasia by high-resolution impedance manometry (HRIM) were recruited prospectively. Patients with refractory GERD symptoms, defined as inadequate symptom response despite eight weeks of proton-pump inhibitor (PPI) therapy but with normal motility on HRIM, were enrolled as the control group. All patients underwent a comprehensive evaluation, including validated symptom questionnaires, EGD, and HRIM. All patients had a biopsy of the esophageal mucosa taken during EGD.

### 2.2. Symptom Evaluation

We assessed patients’ symptom severity with validated symptom questionnaires. The reflux disease questionnaire (RDQ) assesses the frequency and severity of upper gastrointestinal symptoms and covers three main domains: dyspepsia, regurgitation, and heartburn [16]. The Eckardt score assesses the severity of patients with non-obstructive dysphagia and consists of the sum of symptom scores for dysphagia, regurgitation, retrosternal pain, and weight loss [17]. 

### 2.3. Esophageal HRIM

After an overnight fast, all patients underwent an esophageal HRIM. The examination was performed with a water-perfused (4.2 mm in diameter, with 22 closely spaced pressure sensors at 1 cm intervals and 12 impedance channels at 2 cm intervals) or solid-state system (3.3 mm in diameter, with 36 closely spaced pressure sensors at 1 cm interval and 16 impedance channels at 2 cm intervals) (MMS, Medical Measurements Systems, Enschede, The Netherlands). The manometric signals were recorded at a frequency of 20 Hz and stored. The data was analyzed by the package analysis software from MMS (MMS, Medical Measurements Systems, Enschede, The Netherlands) and confirmed by an experienced gastroenterologist. The diagnoses of motility disorders were made according to the criteria stipulated by the Chicago Classification v3.0 [18]. The diagnosis of achalasia is based on absent peristalsis and impaired LES relaxation demonstrated by HRIM [18]. We further classified achalasia into the following three subtypes based on the manometric patterns: type I achalasia presents with no significant pressurization within the esophageal lumen; type II achalasia presents with at least 20% pan esophageal pressurization; and type III achalasia has premature or spastic distal esophageal contractions [19].

### 2.4. Upper Endoscopy and Esophageal Mucosal Biopsy

After patients fasted for at least eight hours, an EGD was performed by experienced endoscopists. Upon endoscopic insertion, we first determined the presence of any solid or liquid retention inside the esophageal lumen. The distal esophagus and the esophagogastric junction (EGJ) were carefully evaluated for the presence of any erosions (i.e., erosive esophagitis) or hiatal hernia. Hiatal hernia was defined as a distance of at least 2 cm between the top of the gastric folds and the diaphragmatic hiatus. Absence of erosions and hiatal hernia was defined as normal EGJ morphology. Then we assessed the resistance over the EGJ while advancing the endoscope into the stomach. An endoscopic biopsy was performed with standard forceps at the mid esophagus for the histologic evaluation of the esophageal mucosa. Each specimen was measured to be approximately 0.3 cm × 0.3 cm × 0.4 cm.

### 2.5. Histopathologic Evaluation of the Esophageal Mucosa

The specimens were fixed with 10% formalin for 24 h, dehydrated with ethanol and xylene, embedded with paraffin, and then sliced into 4 μm sections for examination. Hematoxylin and eosin staining (H&E) was performed for the histological images. Conventional histopathological characteristics of GERD consist of basal layer hyperplasia, elongation of the lamina propria papillae, intra-epithelial infiltration of inflammatory cells, dilated intercellular spaces, and erosions [12,13]. Previously reported histopathology characteristics of achalasia in the literature include the absence of myenteric ganglion cells, atrophy of the muscularis mucosae (MM), a wavy epithelial pattern, and fibrosis [20]. We reorganized the aforementioned histological findings and classified them into four structured criteria, including two conventional histological findings of esophagitis and two novel histological criteria (Table 1 and Figure 1): 1. Basal cell hyperplasia or papillae elongation (Figure 1a): increased basal cell layer to more than 15% of total thickness of squamous epithelium or papillae extending into the upper third of the epithelium [21]; 2. Eosinophilic infiltration (Figure 1b): presence and confirmation of at least one or more intraepithelial eosinophils per high-power field (HPF) [20]; 3. Petechiae formation (Figure 1c): the presence of extravasation of red blood cells (≥1 per HPF) from the capillaries in lamina propria papillae; and 4. Hypertrophy of the MM (Figure 1d): non-interruption and evident thickening of the smooth muscle bundle of the MM [22,23]. Histopathological evaluation of all the specimens was performed by an experienced gastrointestinal pathologist, who was blinded to the participants’ clinical diagnosis.

### 2.6. Statistical Analysis

The recorded continuous and categorical (non-continuous) variables were presented as mean ± standard deviation and numbers (%). Student’s *t*-tests were conducted to examine the difference of baseline characteristics and symptom severity scores between the achalasia and control groups. Chi-squared tests were applied to compare the histopathologic findings between the achalasia and control group and the different subtypes of achalasia. To identify significant predictors for achalasia, we used the logistic regression model and expressed the results as the odds ratio (OR) with the corresponding 95% confidence intervals (CI). The predictors included various clinical, endoscopic, and histopathologic characteristics. In Model 1, we included the predictors of Eckardt score and endoscopic findings, and in Models 2, we added the histopathologic features in prediction. The area under the receiver operating characteristic (AUROC) curve provided a measure of the discriminative power before and after adding the histopathologic features in prediction models. Calibration was tested with the Hosmer–Lemeshow test for goodness of fit of the model’s prediction. All statistical analyses were performed using SPSS 18.0 (SPSS, Inc., Chicago, IL, USA) and Stata 14 software (StataCorp LLC, College Station, TX, USA). All *p*-values were two-sided, with *p* < 0.05 considered statistically significant.

## 3. Results

### 3.1. Patient Characteristics

From July 2015 to March 2020, 54 patients with achalasia (18 males (33.3%), mean 52.9 ± 14.7 years) and 46 patients with refractory GERD symptoms and normal motility on HRIM (13 males (28.3%), 51.9 ± 12.5 years) were enrolled in this study. There was no discrepancy in age, sex, BMI, and waist circumference between the achalasia and the control group. The mean 4-s integrated relaxation pressure (IRP 4s) on HRM was 23.7 ± 12.4 mmHg in the achalasia group and 7.7 ± 4.2 mmHg in the control group (*p* < 0.001). Regarding the symptom severity scores (Table 2), there was no group difference on the RDQ score, but the Eckardt score was significantly higher in achalasia patients (5.1 ± 2.3 vs. 3.1 ± 1.8, *p* < 0.001) due to more severe symptoms of dysphagia, body weight loss, and regurgitation. There was more erosive esophagitis (21.7% vs. 7.4%, *p* = 0.04) and hiatal hernia (15.2% vs. 0%, *p* < 0.001) on endoscopic examination in the GERD group. Food retention in the esophageal lumen was only present in the achalasia group (57.4% vs. 0%, *p* < 0.001).

### 3.2. Histopathological Comparison of the Esophageal Mucosa between Achalasia and Refractory GERD

The comparison of the histopathologic patterns in the four formulated criteria is shown in Table 1. Among the 54 patients with achalasia, basal cell hyperplasia or papillae elongation, eosinophilic infiltration, petechiae formation, and hypertrophy of the MM were observed in 53 (98.1%), 13 (24.1%), 47 (87.0%), and 25 (46.3%) cases, respectively. Compared with the control group (Figure 2), the achalasia group had a higher prevalence of petechiae formation (87% vs. 69.6%, *p* = 0.033) and hypertrophy of the MM (46.3% vs. 21.7%, *p* = 0.01). The coexistence of petechiae formation and hypertrophy of the MM was significantly higher in achalasia patients (40.7% vs. 15.2%, *p* = 0.005). The prevalence of basal cell hyperplasia and papillae elongation were high in both groups without significant differences (98.1% vs. 93.5%, *p* = 0.235).

### 3.3. Histopathological Comparison of the Esophageal Mucosa between the Different Subtypes of Achalasia 

The most common subtype of achalasia encountered was type II (*n* = 28), followed by type I (*n* = 24) and type III (*n* = 2). The histopathology of the mucosal biopsies showed no differences between the three subtypes (Table 3). We further compared the two most common subtypes of achalasia. There was no group difference in the RDQ and the Eckardt scores between type I and type II achalasia, but type I achalasia more commonly manifested with esophageal food retention when compared to type II (83.3% vs. 35.7%, *p* = 0.001), as shown in Appendix A. Type I achalasia was further classified into sigmoid-type and non-sigmoid-type according to a timed barium esophagogram [24]. Hypertrophy of the MM was less frequently observed (33.3% vs. 75.0%, *p* = 0.041) in patients with type I achalasia (12, 50%) who presented with sigmoid-type lumen (Appendix A).

### 3.4. Regression Analyses for Prediction of Achalasia

The results of the univariable and multivariable logistic regression modeling are shown in Table 4. Univariate analyses revealed that Eckardt score ≥ 4 (OR = 4.88; 95% CI: 2.08–11.41), normal EGJ morphology or esophageal food retention (OR = 13.87; 95% CI: 2.98–64.5), petechiae formation (OR = 2.94; 95% CI: 1.07–8.09), and hypertrophy of the MM (OR = 3.10; 95% CI: 1.29–7.49) were significant predictors for achalasia. Multivariable analyses also confirmed these were independent predictors for achalasia. We further assessed the accuracy of the prediction models and compared by AUROC curve analysis. As shown in Figure 3, the performance of Model 2 in the prediction of achalasia improved significantly after including the histological features into the Model 1 (0.793 vs. 0.758, *p* = 0.033).

## 4. Discussion

This study assessed the histopathological findings from esophageal mucosal biopsies using two well-documented histological characteristics of esophagitis (basal cell hyperplasia or papillae elongation and eosinophilic infiltration) and two novel histological characteristics (petechiae formation and hypertrophy of the MM) to differentiate achalasia from refractory GERD. Both groups had a high prevalence of basal cell hyperplasia and papillae elongation, which is the conventional histological evidence of esophagitis [25]. The mechanism of microscopic esophagitis in GERD is mainly related to acid reflux. In contrast, esophagitis in achalasia is not solely related to acidic or food reflux but is more probably mediated by cytokines [26,27].

In this study, eosinophilic infiltration was found in approximately 20–25% of the achalasia and refractory GERD patients without a significant difference between the groups. Eosinophils are normally present throughout the lamina propria of the gastrointestinal tract but are not intra-epithelial [28]. Intraepithelial eosinophilic infiltration is a non-specific finding of chronic esophagitis due to prolonged acid or food reflux and may lead to tissue damage and edema mediated by cytotoxic chemicals [29]. In GERD patients with unexplained dysphagia, Ayazi et al., found that the number of intraepithelial eosinophils correlated with dysphagia severity. Eosinophils secrete products which antagonize muscarinic M2 receptors and cause vagal dysregulation and decreased esophageal muscle contractility, and this may be the probable mechanism of dysphagia in GERD patients [30]. In achalasia patients, neurotoxic secretory products released from eosinophils in the epithelium and the myenteric plexus may play a role in disease pathogenesis [31]. Pro-fibrotic products secreted from eosinophils may result in tissue remodeling and ultimately fibrosis in advanced achalasia [31].

The most significant histopathological differences between the achalasia and GERD groups in the present study were petechiae formation and hypertrophy of the MM. Patients with achalasia had a higher prevalence of petechiae formation and hypertrophy of the MM and the coexistence of both histopathologic features. Food or fluid retention in the esophageal lumen and poorly relaxed LES results in chronic elevated intraluminal pressure, the mechanical stress may cause extravasation of red blood cells from the capillaries in the lamina propria papillae (petechiae formation) [32]. Inhibitory neuronal signals such as nitric oxide suppress smooth muscle proliferation [33]. Loss of inhibitory neurons and nitric oxide secretion in achalasia may lead to muscular hypertrophy, including the MM and the muscularis propria [34,35].

Sato et al., analyzed full-layer mucosal histology in patients with achalasia who received peroral endoscopic myotomy (POEM) and found a correlation of epithelial waves in endoscopic “pinstripe pattern”-positive achalasia [20]. Histological findings of esophagitis such as inflammatory cell infiltration and dilated intercellular spaces were more common in patients with achalasia than in patients diagnosed with esophageal cancer. Additionally, their study showed more atrophy of the MM in advanced achalasia, which is consistent with our findings showing less hypertrophy of the MM in type I achalasia with a sigmoid-type lumen in the subgroup analysis. We hypothesized that hypertrophy of the MM was not evident at the initial phase of the disease. As the disease progresses, the loss of the inhibitory neurons that release nitric oxide results in hypertrophy of the MM. However, patients with the most advanced disease (sigmoid-type or end-stage achalasia) may decompensate and present with atrophy of the MM due to chronic inflammation and fibrotic change [36]. 

Aperistalsis and impaired LES relaxation on high-resolution manometry is the gold standard for the diagnosis of achalasia. However, there are a subset of achalasia patients with typical dysphagia symptoms but relatively preserved LES relaxation presenting with normal IRP, in which further provocative tests such as a rapid drink challenge are necessary to confirm the diagnosis [37]. Early-stage achalasia also lack simple and typical endoscopic features to assist diagnosis. In this study, we have constructed regression models for the prediction of achalasia based on the clinical symptomatology and specific endoscopic findings. After including the histopathologic features into the model, there was a significant improvement in the predictive performance for achalasia. Our study results suggest that the combination of symptom score, endoscopic features, and esophageal biopsy with a dedicated histopathological evaluation may serve as a useful adjunctive diagnostic modality to facilitate in early identification of achalasia. 

This study’s strengths included a prospective study design with a standard protocol including both subjective and objective evaluations to characterize patients with achalasia and patients with refractory GERD. Moreover, a biopsy of the esophageal mucosa is readily accessible during endoscopic evaluation for non-obstructive dysphagia or refractory reflux symptoms. To the best of our knowledge, such delicate analyses of the histopathology of the esophageal mucosa patterns in HRM-confirmed achalasia and its subtypes have not been previously reported. This is also the first study trying to develop an effective predictive model for achalasia based on the combined clinical, endoscopic, and histopathologic characteristics. However, this study has several limitations. First, the sample size was relatively small, especially for type III achalasia. Since achalasia is a rare esophageal motility disorder, further studies with more cases are needed to confirm our findings. Second, immunohistochemistry and special stains were not routinely performed during histopathologic examination. However, the H&E stain is the most widely used and accessible staining method for pathological diagnosis. Moreover, there is still a possibility of the simultaneous presence of esophageal motility disorders and reflux disease, thus making a clear and prompt diagnosis difficult. Therefore, long-term follow-up of this refractory GERD cohort is crucial for a better understanding of achalasia’s complex pathophysiology.

## 5. Conclusions

In conclusion, this study showed a significantly higher prevalence of petechiae formation and hypertrophy of the MM in the achalasia group. These findings suggest that histopathology of the esophageal mucosa obtained via endoscopic biopsy not only help to clarify the underlying pathophysiology of achalasia but may also be adopted as an adjunctive diagnostic modality for achalasia. In addition, we found that combination of clinical, endoscopic, and histopathologic characteristics provided a useful and accurate predictive model of achalasia. Early identification and interventions for patients with achalasia could bring better outcomes and prevent disease progression into end-stage achalasia.

## Figures and Tables

**Figure 1 diagnostics-11-02347-f001:**
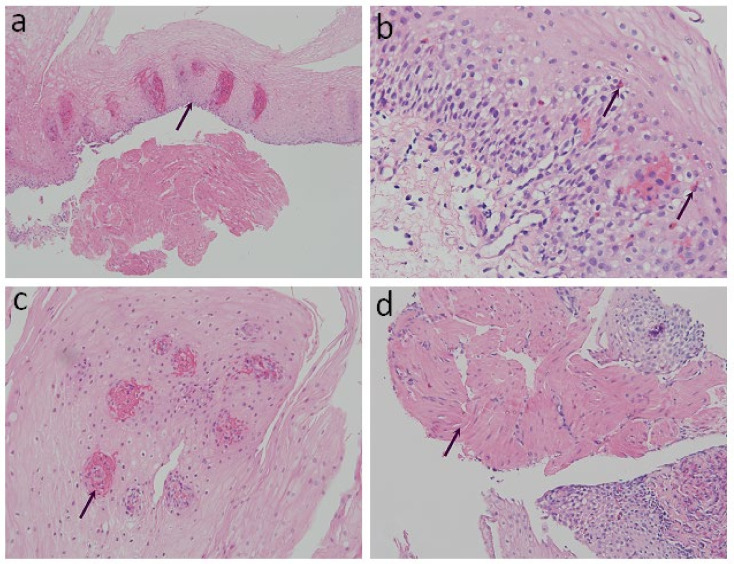
Histopathological findings on esophageal mucosa biopsies. (arrow) (H&E stain): (**a**) Basal cell hyperplasia or papillae elongation (100× magnification); (**b**) Eosinophilic infiltration (200× magnification); (**c**) Petechiae formation (400× magnification); (**d**) Hypertrophy of the muscularis mucosae (200× magnification).

**Figure 2 diagnostics-11-02347-f002:**
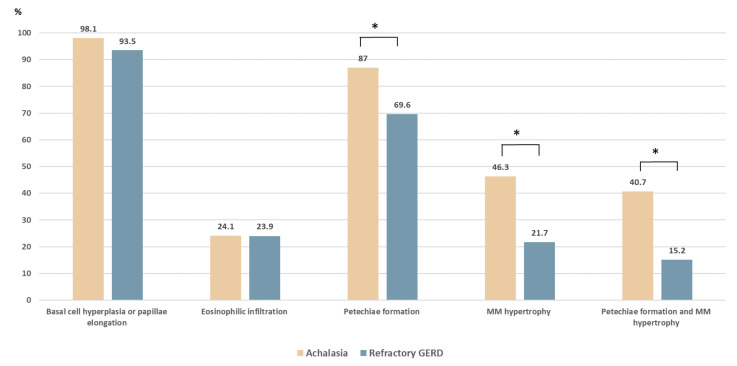
Histopathologic comparison between patients with achalasia and with refractory gastroesophageal reflux disease (GERD). Compared with the refractory GERD group, patients with achalasia had a higher prevalence of hypertrophy of the muscularis mucosae (46.3% vs. 21.7%, *p* = 0.01) and petechiae formation (87.0% vs. 69.6%, *p* = 0.033). The coexistence of petechiae formation and hypertrophy of the muscularis mucosae was significantly higher in patients with achalasia (40.7% vs. 15.2%, *p* = 0.005). * *p* < 0.05.

**Figure 3 diagnostics-11-02347-f003:**
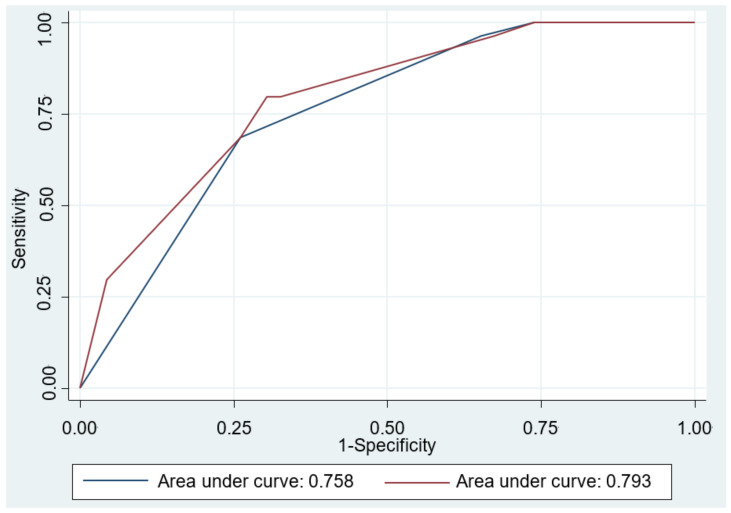
Area under the receiver operating characteristic (AUROC) curve for the predictive model differentiating achalasia from refractory gastroesophageal reflux disease. The AUROC curve was 0.758 in Model 1 (blue line), including Eckardt score ≥ 4, and normal esophagogastric junction morphology or esophageal food retention by esophagogastroduodenoscopy. The AUROC curve was 0.793 after including the histopathologic feature (coexistence of petechiae formation and hypertrophy of the muscularis mucosae) in Model 2 (red line). Comparison between two prediction models showed significant difference (*p* = 0.033).

**Table 1 diagnostics-11-02347-t001:** Definition of histopathologic findings from esophageal mucosa biopsies.

Histopathologic Findings and Definitions	
Basal cell hyperplasia or papillae elongation	Increased basal cell layer to more than 15% of total thickness of squamous epithelium or papillae extending into the upper third of the epithelium.
Eosinophilic infiltration	Presence and confirmation of at least one or more intraepithelial eosinophils per high-power field (HPF).
Petechiae formation	Presence of extravasation of red blood cells (≥1/HPF) from the capillaries in the lamina propria papillae.
Hypertrophy of the muscularis mucosae	Non-interruption (aggregation) and evident thickening of the smooth muscle bundle of the muscularis mucosae.

**Table 2 diagnostics-11-02347-t002:** Clinical characteristics of patients with achalasia and refractory GERD.

	Achalasia	Refractory GERD	*p*-Value
Number of patients	54	46	
Age (years)	52.9 ± 14.7	51.9 ± 12.5	0.344
Male gender (%)	18 (33.3%)	13 (28.3%)	0.585
BMI (kg/m^2^)	21.5 ± 3.6	22.9 ± 3.8	0.066
Waist (cm)	76.4 ± 12.2	79.9 ± 10.6	0.131
**Symptom profile**			
RDQ score	16.5 ± 14.9	19.2 ± 12.9	0.347
Heartburn domain	5.1 ± 6.2	5.8 ± 5.4	0.544
Dyspepsia domain	4.9 ± 6.5	5.0 ± 5.2	0.872
Regurgitation domain	6.5 ± 6.4	8.3 ± 6.2	0.168
Eckardt score	5.1 ± 2.3	3.1 ± 1.8	<0.001
Dysphagia	2.2 ± 0.9	1.1 ± 1.1	<0.001
Retrosternal pain	0.6 ± 0.8	0.7 ± 0.8	0.789
Regurgitation	1.4 ± 1.1	0.8 ± 0.8	0.002
Body weight loss	1.0 ± 1.1	0.5 ± 0.5	0.006
**Endoscopic findings**			
Erosive esophagitis (%)	4 (7.4%)	10 (21.7%)	0.04
Hiatal hernia (%)	0 (0%)	7 (15.2%)	<0.001
Esophageal food retention (%)	31 (57.4%)	0 (0%)	<0.001
**HRIM parameters**			
LES resting pressure (mmHg)	31.8 ± 16.7	19.6 ± 10.3	<0.001
LES IRP 4s (mmHg)	23.7 ± 12.4	7.7 ± 4.2	<0.001
DCI (mmHg∙s∙cm)	-	1541.3 ± 903.2	
Intact peristalsis (%)	0	88.9 ± 16.1	<0.001
Weak peristalsis (%)	0	10.2 ± 15.8	<0.001
Failed peristalsis (%)	100	0.9 ± 2.8	<0.001

Data are presented as mean ± standard deviation or number (percentage). Abbreviations: GERD, gastroesophageal reflux disease; BMI, body mass index; RDQ, reflux disease questionnaire; HRIM, high resolution impedance manometry; LES, lower esophageal sphincter; IRP 4s, integrated relaxation pressure 4s; DCI, distal contractile integral. *p* < 0.05 indicates statistical significance.

**Table 3 diagnostics-11-02347-t003:** Histopathology of achalasia subtypes.

	Type I	Type II	Type III	*p*-Value
Number of patients	24	28	2	
Basal cell hyperplasia or papillae elongation	23 (95.8%)	28 (100%)	2 (100%)	0.544
Eosinophilic infiltration	7 (29.2%)	6 (21.4%)	0 (0%)	0.597
Petechiae formation	21 (87.5%)	24 (85.7%)	2 (100%)	0.849
Hypertrophy of the MM	13 (54.2%)	11 (39.3%)	1 (50%)	0.574

Data are presented as number (percentage). Abbreviations: MM, muscularis mucosae. *p* < 0.05 indicates statistical significance.

**Table 4 diagnostics-11-02347-t004:** Univariable and multivariable logistic regression analyses of achalasia prediction.

Variable	Univariate		Multivariate 1 ^‡^		Multivariate 2 ^§^	
	OR (95% CI)	*p*-Value	aOR (95% CI)	*p*-Value	aOR (95% CI)	*p*-Value
Eckardt score ≥ 4	4.88 (2.08–11.41)	<0.001	4.37 (1.75–10.91)	0.002	4.18 (1.63–10.72)	0.003
Esophageal food retention (EFR)	-	-	-	-	-	-
Normal EGJ morphology ^†^ or EFR	13.87 (2.98–64.5)	0.001	12.18 (2.49–59.58)	0.002	11.59 (2.30–58.37)	0.003
Petechiae formation (PF)	2.94 (1.07–8.09)	0.037	-	-	-	-
Hypertrophy of the muscularis mucosae (HMM)	3.10 (1.29–7.49)	0.012	-	-	-	-
PF or HMM	3.93 (1.16–13.35)	0.028				
Coexistence of PF and HMM	3.83 (1.45–10.11)	0.007	-	-	3.46 (1.14–10.55)	0.029

^‡^ Model 1: Predictors included Eckardt score ≥ 4 and “normal EGJ morphology or esophageal food retention”. Area under ROC curve of model 1 = 0.758. ^§^ Model 2: Predictors included Eckardt score ≥ 4, “normal EGJ morphology or esophageal food retention”, and “coexistence of petechiae formation and hypertrophy of the muscularis mucosae”. Area under ROC curve of Model 2 = 0.793. The *p*-value of Hosmer–Lemeshow test was 0.432. ^†^ Normal esophagogastric junction (EGJ) morphology indicates absence of erosions and hiatal hernia.

## Data Availability

The data presented in this study are available on request from the corresponding author. The data are not publicly available due to privacy.

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
