# Peer review of "Combination of Symptom Profile, Endoscopic Findings, and Esophageal Mucosal Histopathology Helps to Differentiate Achalasia from Refractory Gastroesophageal Reflux Disease"

_diagnostics, 2021, doi:10.3390/diagnostics11122347_

Round 1
Reviewer 1 Report
This is a very good work. The scientific value is high, the study was very well planned, the method is reliable and the hypothesis very important. The analysis is high levelled and the presentation is excellent. The conclusion is well supported by the results.
Author Response
We appreciate your comments. Thank you very much.

Reviewer 2 Report
This is an excellent paper on a medical problem of importance. The manuscript conteins some novelty value and the conduction of this study is scientifically sound. The list of references is adequate and the authors have support for their conclusion in the results of the study.
Author Response

(The authors gave the same response as above.)

Reviewer 3 Report
The paper “Combination of Symptom Profile, Endoscopic Findings, and Esophageal Mucosal Histopathology Helps to Differentiate Achalasia from Refractory Gastroesophageal Reflux Disease” presents novel approach to identify the clinical and histopathologic features that can differentiate achalasia and refractory reflux disease. GERD is a very common disease worldwide, some cases of achalasia might be not diagnosed properly as the symptoms may be similar to those in reflux disease.
In the affiliation section:
In the introduction section please pay more attention to POEM procedure that seems to be crucial treatment right now.
Minor language errors:
Department of pathology- please change to Department of Pathology
Two e-mail addresses have internet link- should be deleted
Please change 24-hour PH to 24-hour pH
2.6. Statistical Analysis bold font should be changed
- Discussion heading font should be corrected
The manuscript proves that endoscopic biopsy may also adopted as an adjunctive diagnostic tool for achalasia
I recommend the manuscript for publication after minor revisions mentioned above.
Author Response
- In the introduction section please pay more attention to POEM procedure that seems to be crucial treatment right now.
Response: Thanks for your suggestion. Accordingly, we have added a paragraph regarding the characteristics of this promising POEM procedure for achalasia treatment in the Introduction section.
- Minor language errors:
Department of pathology- please change to Department of Pathology
Two e-mail addresses have internet link- should be deleted
Please change 24-hour PH to 24-hour pH
Response: Thanks for your kind reminder. We have revised these typos in the revised manuscript.
- 6. Statistical Analysis bold font should be changed
- Discussion heading font should be corrected
Response: Thanks for your kind reminder. We have changed the fonts in the revised manuscript.
- The manuscript proves that endoscopic biopsy may also adopted as an adjunctive diagnostic tool for achalasia. I recommend the manuscript for publication after minor revisions mentioned above.
Response: We appreciate your comments. Thank you very much.
